# Association of the Cumulative Live Birth Rate with the Factors in Assisted Reproductive Technology: A Retrospective Study of 16,583 Women

**DOI:** 10.3390/jcm12020493

**Published:** 2023-01-06

**Authors:** Qiumin Wang, Dan Qi, Lixia Zhang, Jingru Wang, Yanbo Du, Hong Lv, Lei Yan

**Affiliations:** 1Center for Reproductive Medicine, Shandong University, Jinan 250012, China; 2Key Laboratory of Reproductive Endocrinology of Ministry of Education, Shandong University, Jinan 250012, China; 3Shandong Key Laboratory of Reproductive Medicine, Jinan 250012, China; 4Maternal and Child Health and Family Planning Service Center of Yanggu County, Liaocheng 252300, China

**Keywords:** dose–response relationship, cumulative live birth, anti-Müllerian hormone, age, follicle-stimulating hormone, transferred embryos

## Abstract

The cumulative live birth rate (CLBR) can better reflect the overall treatment effect by successive treatments, and continuous rather than categorical variables as exposure variables can increase the statistical power in detecting the potential correlation. Therefore, the dose–response relationships might find an optimal dose for the better CLBR, offering evidence-based references for clinicians. To determine the dose–response relationships of the factors and the optimal ranges of the factors in assisted reproductive technology (ART) associated with a higher CLBR, this study retrospectively analyzed 16,583 patients undergoing the first in vitro fertilization (IVF) or intracytoplasmic sperm injection (ICSI) from January 2017 to January 2019. Our study demonstrated the optimal ranges of age with a higher CLBR were under 32.10 years. We estimated the CLBR tends to increase with increased levels of AMH at AMH levels below 1.482 ng/mL, and the CLBR reaches a slightly high level at AMH levels in the range from 2.58–4.18 ng/mL. The optimal ranges of basal FSH with a higher CLBR were less than 9.13 IU. When the number of cryopreserved embryos was above 1.055 and the number of total transferred embryos was 2, the CLBR was significantly higher. In conclusion, there is a non-linear dose–response relationship between the CLBR with age, AMH, basal FSH, and the number of cryopreserved embryos and total transferred embryos. We proposed the optimal ranges of the five factors that were correlated with a higher CLBR in the first oocyte retrieval cycle, which may help consultation at IVF clinics.

## 1. Introduction

Currently, the incidence of infertility is gradually increasing, which is a global medical concern affecting between 8 and 30% of reproductive-age couples worldwide [1,2,3,4]; hence, the demand for infertility treatment is increasing [5]. Concomitant with the development of assisted reproductive technology (ART), an increasing number of infertility couples obtain a live birth through these technologies. Frozen embryo transfer (FET) is gaining popularity for its advantages of convenience, safety, and efficacy [6,7,8]. The cumulative live birth rate (CLBR) per oocyte retrieval (including fresh embryo transfer and subsequent FET) can better reflect the overall treatment effect by successive treatments [9]. Likewise, the CLBR is an important indicator of common concern for both clinicians and patients.

It has long been known that maternal age is the most significant factor affecting ART outcomes [10]. Previous research found that female obesity adversely affected the CLBR in their first in vitro fertilization (IVF) or intracytoplasmic sperm injection (ICSI) cycles [11]. Women with diminished ovarian reserves had substantially lower live birth rates [12], and the anti-Müllerian hormone (AMH) levels, the basal follicle-stimulating hormone (FSH) levels, and AFC can be used as the indicators for ovarian reserve function [13,14,15]. There are also studies that found an additional predictor of the CLBR was the number of retrieved oocytes [16]. However, most of the previous research treated these factors as categorical variables, which hampered the establishment of clear dose–response relationships between these factors and the CLBR. Moreover, dose–response relationships might find an optimal dose for the better CLBR, offering evidence-based references for clinicians.

Therefore, we designed a retrospective study to determine the optimal ranges of the factors in ART associated with a higher CLBR in women undergoing the first IVF/ICSI.

## 2. Materials and Methods

### 2.1. Study Design and Population

This is a retrospective population-based study. Patients with a total of 30,530 retrieval cycles were collected in this study between January 2017 and January 2019 from the Center for Reproductive Medicine, Shandong University. The following cycles were excluded: all preimplantation genetic testing (PGT); IVF/ICSI with donor oocytes; gamete transfer; oocyte cryopreservation; not the first retrieval cycles; no AMH value or AMH > 1 year from oocyte retrieval; premature ovarian insufficiency (POI) or polycystic ovary syndrome (PCOS). The final sample size for analysis was 16,583. This study was approved by the institutional review board of the Center for Reproductive Medicine, Shandong University (2022–59).

### 2.2. IVF/ICSI Protocols

All patients received a controlled ovarian stimulation protocol, oocyte retrieval, fertilization, an embryo cultured and cryopreserved in vitro, and luteal phase support for fresh embryo transfer (ET), or endometrial preparation and luteal phase support for frozen embryo transfer, according to a routine method [17]. The protocols for controlled ovarian stimulation in our study included luteal phase gonadotropin-releasing hormone (GnRH) agonist long protocols (6536, 39.4%), GnRH agonist short protocols (4974, 30.0%), GnRH antagonist protocols (2842, 17.1%), follicular phase GnRH agonist long protocol (1375, 8.3%), mild stimulation protocol (258, 1.6%), natural cycle protocol (248, 1.5%), GnRH agonist ultrashort protocol (12, 0.1%), and other protocols (338, 2.0%). The IVF-ET protocols used in our center have been described in detail previously [7,18]. Ovarian response was monitored using ultrasonography and serum sex steroid levels. A dose from 4000 to 10,000 IU human chorionic gonadotropin (HCG) was administered when the size of at least two follicles reached 18 mm, and oocyte retrieval was performed from 34 to 36 h later. According to the male partner’s sperm quality, oocytes were fertilized by conventional IVF/ICSI. All embryos were frozen, or up to two fresh embryos were transferred, at the cleavage or blastocyst stage after fertilization. Luteal phase support [18] was started after oocyte retrieval in women with fresh embryos transferred. Frozen blastocysts were thawed and transferred, and subsequently provided luteal phase support protocol according to the different endometrial preparation programs [19]. Pregnancy outcome follow-ups were carried out as previously described [20].

### 2.3. Outcome

Our primary outcome of interest was the CLBR in the first index retrieval cycle. The CLBR was defined as the percentage of live births per patient from the first index retrieval cycle, including all fresh or subsequent two years of frozen embryo transfer after the oocyte retrieval. Live birth (LB) was defined as the delivery of any viable infant at 28 weeks or more of gestation.

### 2.4. Statistical Analysis

All analyses and plotting were performed with IBM SPSS statistics 26.0, R 4.1.3, or GraphPad Prism 9. Continuous numeric variables were expressed as mean ± SD, and categorical variables were described as percentages. The Student’s *t*-test was used for continuous numeric variables and the Chi-squared test was used for categorical variables. Multivariate logistic regression analyses were performed to identify the influencing factors of the CLBR. The candidate variables for multivariate logistic regression analyses were those with *p <* 0.05 after the univariate analyses. Multivariate logistic regression analyses (the backward logistic regression method) were conducted by fitting a logistic regression model. Logistic regression was expressed as an odds ratio (OR) with 95% confidence intervals (CI), and a forest plot was drawn by GraphPad Prism 9. The dose–response relationship between variables (age, AMH, basal FSH, the number of cryopreserved embryos, and total transferred embryos) and the odds ratio of the CLBR was evaluated by a restricted cubic spline (RSC) with covariates adjusted. Sensitivity analyses were performed to evaluate the stability of our findings by restricting the analytic samples to ovulatory women (*n* = 16,474) and women without endometriosis (*n* = 15,605), respectively. *p* < 0.05 was considered statistically significant.

## 3. Results

Of 30,530 retrieval cycles, 16,583 retrieval cycles/patients were included in the final analysis. The study population is presented in a flow chart (Figure 1).

### 3.1. Characteristics of the Study Population

The characteristics of the 16,583 retrieval cycles/patients are shown in Table 1. A total of 7967 (48.04%) patients achieved a live birth. Compared to those who did not obtain a live birth, patients who obtained a live birth were younger (31.00 ± 4.15 vs. 34.221 ± 5.65, *p <* 0.001), had slightly lower BMI (23.67 ± 3.50 vs. 24.11 ± 3.47, *p <* 0.001), slightly shorter duration of infertility (3.62 ± 2.66 vs 3.84 ± 3.24, *p <* 0.001), and slightly lower FBG (5.24 ± 0.81 vs. 5.27 ± 0.79, *p* = 0.028). They had higher AMH levels (3.87 ± 2.75 vs. 2.74 ± 2.64, *p <* 0.001) and lower basal FSH (6.60 ± 1.93 vs. 7.54 ± 3.10, *p <* 0.001), and they were less likely to be parous women (51.30% vs. 62.50%, *p <* 0.001). There were differences in the incidence of uterine factor infertility (14.60% vs. 22.30%, *p <* 0.001), male factor infertility (17.30% vs. 12.60%, *p <* 0.001), and unexplained infertility (5.30% vs. 7.60%, *p <* 0.001) between patients who obtained a live birth or were not among infertility etiologies. Versus those who did not obtain a live birth, women who had a live birth had a lower total Gonadotropin dose (2000.39 ± 969.78 vs. 2211.21 ± 1207.59, *p <* 0.001), a slightly greater number of retrieved oocytes (11.19 ± 5.57 vs 7.69 ± 5.91, *p <* 0.001), more cryopreserved embryos (3.05 ± 2.51 vs. 1.35 ± 2.12, *p <* 0.001) and transferred embryos (1.83 ± 0.80 vs. 1.10 ± 1.14, *p <* 0.001), a higher proportion of cycles with oocytes retrieved (100.00% vs. 96.70%, *p <* 0.001), embryos cryopreserved (86.60% vs. 47.90%, *p <* 0.001), and total transferred embryos (100% vs. 59.30%, *p <* 0.001).

### 3.2. Association of AMH and Other Factors with CLBR

A multivariate analysis was performed by logistic regression (the backward logistic regression method) to examine the association between the CLBR and those candidate variables with *p* < 0.05 after the univariate analyses. Those candidate variables included age, BMI, duration of infertility, FBG, AMH, basal FSH, infertility type, uterine factor infertility, male factor infertility, unexplained infertility, total Gonadotropin dose, the number of retrieved oocytes and cryopreserved embryos, and the number of total transferred embryos. In adjusted models, age (OR 0.910, 95%CI 0.903–0.917, *p* < 0.001), AMH (OR 1.021, 95%CI 1.006–1.037, *p* = 0.005), basal FSH (OR 0.967, 95%CI 0.951–0.983, *p* < 0.001), uterine infertility (OR 0.822, 95%CI 0.747–0.904, *p* < 0.001), male infertility (OR 1.185, 95%CI 1.072–1.309, *p* = 0.001), the number of cryopreserved embryos (OR 1.252, 95%CI 1. 231–1.274, *p* < 0.001), and the number of total transferred embryos (OR 2.033, 95%CI 1.954–2.115, *p* < 0.001) were significantly associated with the CLBR (Table 2 and Figure 2). That is, the CLBR increased with AMH, and the number of cryopreserved embryos and total transferred embryos decreased with age and basal FSH. These couples with male infertility more often had a CLBR, and those women with uterine infertility were significantly less likely to have a CLBR than women without uterine infertility. The other covariates (BMI, duration of infertility, FBG, infertility type, unexplained infertility, total Gonadotropin dose, and the number of retrieved oocytes) were unrelated to the CLBR and were not included in the multivariable-adjusted logistic regression model. In sensitivity analyses, the association of these factors with the CLBR was similar to the results in samples of women who were ovulatory (*n* = 16,474) or women without endometriosis (*n* = 15,605), respectively (Table 2).

We further evaluated the dose–response relationship between the CLBR and AMH and other variables (age, basal FSH, and the number of cryopreserved embryos and transferred embryos) by a restricted cubic spline (RSC). The results of the RSC are presented in Figure 3. After adjusted AMH, basal FSH, and the number of cryopreserved embryos and transferred embryos, the CLBR decreased significantly with increasing age (non-linear, *p* < 0.001, Figure 3A). We found the CLBR with an inverse association above 32.10 years of age (OR 0.997, 95%CI 0.994–0.999). The association was more pronounced with increasing age, the OR of the CLBR was 0.016 (95%CI 0.010–0.025) when the age was 48 years. After adjusted age, basal FSH, and the number of cryopreserved embryos and transferred embryos, the CLBR increased significantly with increasing AMH levels (non-linear, *p* < 0.001, Figure 3B). We estimated the CLBR tends to increase with an increase in the levels of AMH at AMH levels below 1.482 ng/mL (OR 0.896, 95%CI 0.817–0.984) and the CLBR to reach a slightly high level at AMH levels in the range from 2.58–4.18 ng/mL (OR > 1.0, however, the 95%CI included 1.0). After adjusted age, AMH, and the number of cryopreserved embryos and transferred embryos, the CLBR was associated with basal FSH (non-linear, *p* = 0.002, Figure 3C). The CLBR was significantly decreased when basal FSH was more than 9.13 IU (OR 0.890, 95%CI 0.792–0.999). After adjusted age, basal FSH, AMH, and the number of total transferred embryos, the CLBR was positively correlated with the number of cryopreserved embryos (non-linear, *p* < 0.001, Figure 3D). When the number of cryopreserved embryos was above 1.055 (OR 1.038, 95%CI 1.035–1.041), the CLBR was significantly higher. After adjusted age, basal FSH, AMH, and the number of cryopreserved embryos, the CLBR was associated with the number of total transferred embryos (non-linear, *p <* 0.001, Figure 3E). The CLBR was significantly higher when the number of total transferred embryos was two (OR 1.252, 95%CI 1.151–1.361), and was decreased when the number of total transferred embryos was more than three (OR 0.595, 95%CI 0.535–0.662). When the number of total transferred embryos was eight, the OR of CLBR was only 0.077 (95%CI 0.047–0.127).

In sensitivity analyses, the dose–response relationships between the CLBR and variables (age, AMH, basal FSH, the number of cryopreserved embryos, and the number of total transferred embryos) were similar to the above findings in ovulatory women or women without endometriosis (Appendix A).

## 4. Discussion

In this study, the CLBR was non-linearly associated with age, AMH, basal FSH, and the number of cryopreserved embryos and total transferred embryos. For the first time, we found the optimal ranges of the five factors that were correlated with a higher CLBR in the first oocyte retrieval cycle. The optimal ranges of age with a higher CLBR were under 32 years, the optimal ranges of AMH were from 2.58–4.18 ng/mL (but the differences were not significant), and the optimal ranges of basal FSH were less than 9.13 IU. When the number of cryopreserved embryos was above one and the number of total transferred embryos was two, the CLBR was significantly higher.

Considering the development of modern ART, the CLBR per oocyte retrieval cycle (including fresh embryos transferred and all frozen embryos transferred after oocyte retrieval) has become a more meaningful outcome for patients and clinicians [9]. Therefore, the CLBR served as the main outcome parameter in our study. Previous evidence has demonstrated that female age is a major factor influencing fertility [21] and that decreasing fecundity is associated with increasing age, due to a reduced ovarian reserve, poor quality of oocytes, and an increased incidence of embryonic aneuploidy [22,23]. Whether after assisted reproduction or not, the CLBR gradually decreased with an increase in the age of females [24]. The majority of studies [25,26,27,28,29] have reported that the CLBR decreases in females after the age of 35 compared with females under the age of 35; however, in our study, the age moved to 32 years. This result is consistent with the previous opinion [30]. The advanced upper limit of optimal reproductive age might be related to the increase in work stress, bad lifestyle, or environmental pollution today.

AMH is secreted by granulosa cells (GCs) of pre-antral and small antral follicles in the ovary [31]. AMH levels can reflect the number of pre-antral and small antral follicles [32]. Converging evidence revealed that AMH can be considered a reliable indicator of ovarian reserves [33,34] and for predicting the success rate of in vitro fertilization (IVF) [35,36,37]. Recently, a retrospective study with a large sample size pointed out that AMH highly correlates with the CLBR in women with diminished ovarian reserves (DOR) independent of age [14]. The similar result was found in a population of elderly women [38]. Low AMH levels have been reported to have a negative effect on live birth in women undergoing ART [29,39,40,41]. In our study, AMH levels below 1.482 n/mL were the risk factor of the CLBR, which indicated that a low AMH level is associated with poor ovarian reserves. The women with low AMH would yield fewer follicles, fewer transferable embryos, and, therefore, have fewer chances of transfer. AMH with specific ranges would provide better estimates of IVF outcomes [42]. At present, the optimal range of AMH levels has not been clarified. Previous research has found that serum levels of AMH above 3.5 ng/mL do not significantly increase the chance of a live birth [28], which is similar to our study findings in all subjects. Results of a recent study showed that the CLBR had a decreasing trend or was not significantly changed when the serum level of AMH was over 5 ng/mL in young women (under 35 years of age) or over 7 ng/mL in older women (above 35 years). Moreover, the optimal range of AMH levels was reported to be between 5 and 7 ng/mL in all women. [28]. However, the AMH range in that study was artificially set and this way is prone to anthropogenic impact. In the current analysis, RCS models allowed for departures from linearity, which can flexibly model the relationship of AMH and CLBR to determine the optimal range of AMH levels. Moreover, we used continuous rather than categorical variables, as exposure variables can increase the statistical power in detecting the potential correlation [43].

A previous study has concluded that high AMH levels might indicate more quantity of oocytes or embryos, rather than a higher quality of oocytes or embryos [44]. However, the specific mechanism of the high AMH levels affecting the quality of oocytes has not been clarified. There was no correlation between AMH levels with oocyte quality in women of advanced age [45]. Therefore, the CLBR is not significantly improved in the present study when the serum AMH level is above 4.18 ng/mL. It was speculated that a high AMH level might be associated with adverse perinatal outcomes. Recently, research found that high levels of serum AMH had a significantly higher risk of miscarriage in women with or without PCOS [46]. Similarly, an increased rate of miscarriage has been reported in women with high or low AMH levels [47]. High serum AMH levels were also associated with an increased risk of preterm delivery in women with PCOS [48]. The result was similar to those from another study, and another study proposed that closer monitoring during the third trimester in patients with serum levels of AMH over 9.3 ng/mL might be required [49]. In addition, we excluded the possible interference of the women with PCOS or POI to the results, and adjusted other potentially influencing factors such as age, basal FSH, or the number of cryopreserved embryos and total transferred embryos.

The basal FSH level is another clinical index to evaluate ovarian reserve, as basal FSH concentration increases when the ovarian reserve declines [50]. Compared with AMH, the ability of basal FSH to predict pregnancy outcomes was poor [35,51,52]. Our results showed that basal FSH was higher than 9.13 IU, and the CLBR was significantly decreased. Compared to previously published results, the cut-off of FSH in our study was a lower level, and a more accurate range could lead to more valuable information for the clinician. This could be useful for clinicians in clinical decision-making about ART for these patients. Over the past few decades, the number of FETs has continuously increased for the improvement of embryo culture conditions and the development of vitrification techniques [53]. FET is gaining popularity for its advantages of convenience, safety, and efficacy [6,7,8]. Our results were similar to those of the previous study [54,55,56], and the CLBR was positively correlated with the number of cryopreserved embryos. Fewer cryopreserved embryos reduces the chance to transfer; however, in our study, the CLBR no longer increased, as the number of cryopreserved embryos increased to a certain number. It might be that the top-quality embryos were given priority for transfer, so that transferring the surplus of poor embryos would not have significantly helped to increase the CLBR [57]. The same patients with several embryo transfers were unsuccessful, which may be associated with other factors, such as the patients with a thin endometrium. Further, we identified that the CLBR was significantly higher as the number of totals transferred was two. This result is different from the findings of previous research [58]. This could be due to high-quality embryos being preferentially selected for transfer, and women without obtained live birth needed to perform repeated embryo transfers. It is an important reminder for clinicians not to pursue the number of transfers, and prepared endometrium is an important factor affecting the success of ART when top-quality embryos are transferred [59].

There are strengths and weaknesses in the present study. First, the CLBR served as the main outcome parameter in our study. CLBR per oocyte retrieval can better reflect the overall treatment effect by successive treatments, and it has become a more meaningful outcome for patients and clinicians. Second, in this single-center and large-scale study, the embryos were cultured in the same laboratory conditions, which minimized potential bias to a large extent. Third, RCS models in the current analysis allowed for departures from linearity, which can flexibly model the relationship between these factors and the CLBR to determine the optimal ranges of these factors. We used continuous rather than categorical variables as exposure variables can increase the statistical power in detecting the potential correlation [43]. Dose–response relationships could find an optimal dose for the better CLBR, and we surmise that the results obtained in the present study are more precise and clinically more efficient than the previous research. In addition, we excluded the possible interference of the women with PCOS or POI from the results and adjusted other potentially influencing factors. However, the main limitation of our study was its retrospective nature; although, we attempted to mitigate these limitations by controlling for confounders and performing sensitivity analyses. Future prospective multicenter studies are required to verify our findings. Another limitation of our study was the lack of data on other risk factors, such as maternal lifestyle habits. Future studies should evaluate other potential confounders, such as work stress, lifestyle habits, or environmental pollution.

## 5. Conclusions

In conclusion, our study suggested that there is a non-linear dose–response relationship between the CLBR with age, AMH, basal FSH, and the number of cryopreserved embryos and total transferred embryos. We found the optimal ranges of the five factors that were correlated with a higher CLBR in the first oocyte retrieval cycle, which may provide a scientific basis for the clinical management and treatment of IVF. The results could guide clinicians to better manage the ART procedures and provide proper counseling services for infertile couples.

## Figures and Tables

**Figure 1 jcm-12-00493-f001:**
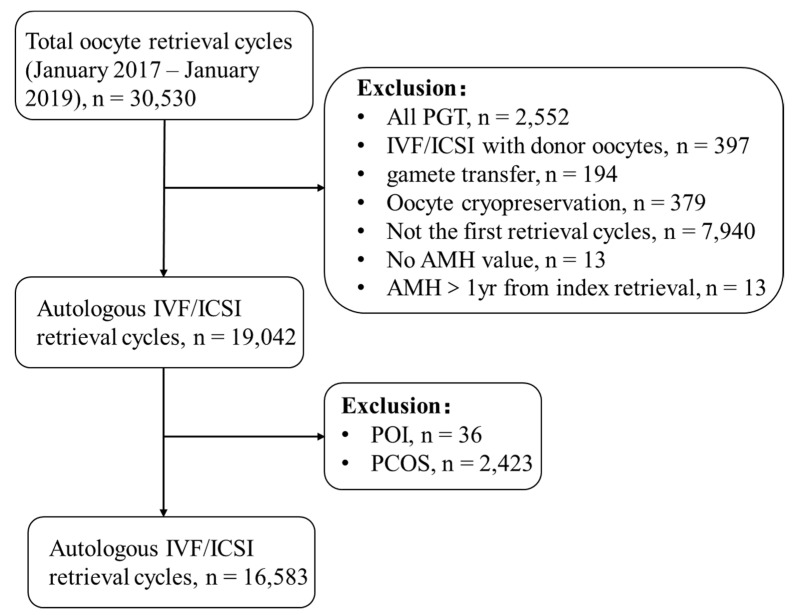
Flow chart of the study population. PGT: preimplantation genetic testing; IVF: in vitro fertilization; ICSI: intracytoplasmic sperm injection; AMH: anti-Müllerian hormone; POI: premature ovarian insufficiency; PCOS: polycystic ovary syndrome.

**Figure 2 jcm-12-00493-f002:**
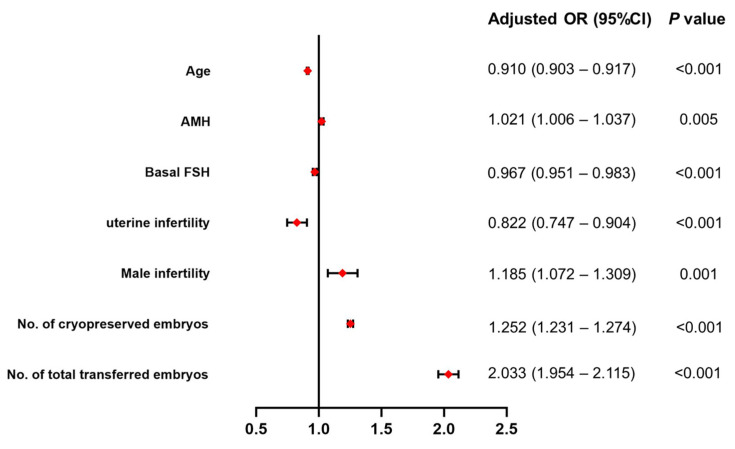
Forest plot of multivariable regression analyses. AMH: anti-Müllerian hormone; FSH: follicle-stimulating hormone. The red square is adjusted OR.

**Figure 3 jcm-12-00493-f003:**
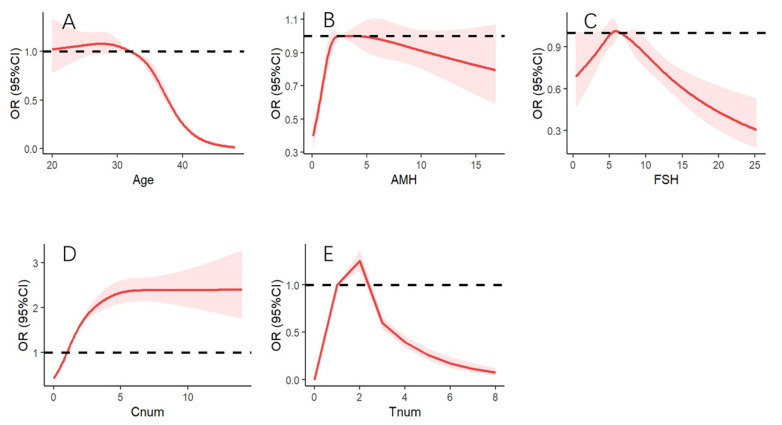
Association of the five factors with CLBR ((**A**). Association of age with CLBR; (**B**). Association of AMH with CLBR; (**C**). Association of FSH with CLBR; (**D**). Association of the number of cryopreserved embryos with CLBR; (**E**). Association of the number of total transferred embryos with CLBR). AMH: anti-Müllerian hormone; FSH: follicle-stimulating hormone; Cnum: number of cryopreserved embryos; Tnum: number of total transferred embryos.

**Table 1 jcm-12-00493-t001:** Baseline and stimulation cycle characteristics.

Characteristic	Total (*n* = 16,583)	No Live Birth (*n* = 8616)	Live Birth (*n* = 7967)	*p* Value
Age	32.68 ± 5.24	34.22 ± 5.65	31.00 ± 4.15	<0.001
BMI	23.90 ± 3.49	24.11 ± 3.47	23.67 ± 3.50	<0.001
Duration of infertility	3.73 ± 2.98	3.84 ± 3.24	3.62 ± 2.66	<0.001
FBG	5.25 ± 0.80	5.27 ± 0.79	5.24 ± 0.81	0.028
AMH	3.28 ± 2.75	2.74 ± 2.64	3.87 ± 2.75	<0.001
Basal FSH	7.09 ± 2.64	7.54 ± 3.10	6.60 ± 1.93	<0.001
Gravidity ≥ 1 (%)	57.20	62.50	51.30	<0.001
Infertility etiology (%)				
Tubal factor	79.40	79.10	79.70	0.327
Uterine factor	18.60	22.30	14.60	<0.001
Male factor	14.80	12.60	17.30	<0.001
Unexplained	6.50	7.60	5.30	<0.001
Endometriosis	5.90	6.10	5.70	0.276
Ovulatory dysfunction	0.70	0.60	0.70	0.502
Total Gonadotropin dose (IU)	2109.92 ± 1104.77	2211.21 ± 1207.59	2000.39 ± 969.78	<0.001
No. of retrieved oocytes	9.37 ± 6.01	7.69 ± 5.91	11.19 ± 5.57	<0.001
Cycles with oocytes retrieved (%)	98.30	96.70	100.00	<0.001
No. of cryopreserved embryos	2.17 ± 2.47	1.35 ± 2.12	3.05 ± 2.51	<0.001
Cycles with embryos cryopreserved (%)	66.50	47.90	86.60	<0.001
No. of total transferred embryos	1.45 ± 1.06	1.10 ± 1.14	1.83 ± 0.80	<0.001
Cycles with transferred embryos (%)	78.80	59.30	100.00	<0.001

BMI: body mass index; FBG: fasting blood glucose; AMH: anti-Müllerian hormone; FSH: follicle-stimulating hormone.

**Table 2 jcm-12-00493-t002:** Multiple logistic regression analysis.

Parameter	Adjusted OR (95%CI)	*p* Value
Total population (*n* = 16,583)		
Age	0.910 (0.903–0.917)	<0.001
AMH	1.021 (1.006–1.037)	0.005
Basal FSH	0.967 (0.951–0.983)	<0.001
Uterine infertility	0.822 (0.747–0.904)	<0.001
Male infertility	1.185 (1.072–1.309)	0.001
No. of cryopreserved embryos	1.252 (1.231–1.274)	<0.001
No. of total transferred embryos	2.033 (1.954–2.115)	<0.001
Ovulatory Women (*n* = 16,474)		
Age	0.909 (0.902–0.917)	<0.001
AMH	1.023 (1.007–1.038)	0.003
Basal FSH	0.966 (0.950–0.982)	<0.001
Uterine infertility	0.824 (0.748–0.907)	<0.001
Male infertility	1.181 (1.069–1.306)	0.001
No. of cryopreserved embryos	1.252 (1.231–1.274)	<0.001
No. of total transferred embryos	2.035 (1.955–2.117)	<0.001
Women without endometriosis (*n* = 15,605)		
Age	0.910 (0.902–0.917)	<0.001
AMH	1.020 (1.004–1.036)	0.012
Basal FSH	0.968 (0.951–0.985)	<0.001
Uterine infertility	0.786 (0.711–0.869)	<0.001
Male infertility	1.175 (1.061–1.300)	0.002
Total Gonadotropin dose/100	0.997 (0.993–1.000)	0.090
No. of cryopreserved embryos	1.247 (1.225–1.269)	<0.001
No. of total transferred embryos	2.027 (1.946–2.112)	<0.001

AMH: anti-Müllerian hormone; FSH: follicle-stimulating hormone.

## Data Availability

The data underlying this article will be shared upon reasonable request to the first or corresponding authors.

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
