# Peer review of "Association of the Cumulative Live Birth Rate with the Factors in Assisted Reproductive Technology: A Retrospective Study of 16,583 Women"

_jcm, 2023, doi:10.3390/jcm12020493_

Round 1

Reviewer 1 Report

This is a quite interesting study evaluating the relevance of different factors on the cumulative live birth rate in in vitro fertilization programs. However, the manuscript requires deep revision since the clinical relevance of the results obtained are not clear. Moreover, the document has several grammatical and typographical errors that must be corrected.

Title: “other factors”. This term is very ambiguous, so I strongly recommend the authors to revise and improve the title.

Line 14. “To determine the optimal ranges of these factors associated with a higher CLBR”. Please, specify which are these factors. Moreover, authors must indicate the complete name of CLBR to improve the understanding of the sentence.

Line 55. “30530”. Please, correct as “30,530”.

Line 61. “16583”. Please, correct as “16,583”. Please, revise the whole document and correct accordingly.

Figure 1. Authors must improve the figure legend, by indicating the meaning of the initials.

Table 1. Please, revise and improve the format of this table. Authors must also provide and appropriate table legend.

Table 2. Authors must provide and appropriate table legend.

Figures 2 and 3. Authors must provide and appropriate legend for both figures.

The Discussion section needs to be improved by highlighting the physiological relevance of the results obtained and their implications in clinical practice.

The Conclusion section also needs deep revision and improvement by highlighting most significant results.

Author Response

This is a quite interesting study evaluating the relevance of different factors on the cumulative live birth rate in in vitro fertilization programs. However, the manuscript requires deep revision since the clinical relevance of the results obtained are not clear. Moreover, the document has several grammatical and typographical errors that must be corrected.

Title: “other factors”. This term is very ambiguous, so I strongly recommend the authors to revise and improve the title.

Response: Thank you for your advice, the title has been revised accordingly (yellow highlight) in the re-submitted manuscript.

Line 14. “To determine the optimal ranges of these factors associated with a higher CLBR”. Please, specify which are these factors. Moreover, authors must indicate the complete name of CLBR to improve the understanding of the sentence.

Response: Thank you for your advice, this part has been corrected accordingly (yellow highlight) in the re-submitted manuscript. And we have determined these factors in later results according to logistic regression analyses.

Line 55. “30530”. Please, correct as “30,530”.

Response: Thank you for your advice, this part has been corrected accordingly (yellow highlight) in the re-submitted manuscript.

Line 61. “16583”. Please, correct as “16,583”. Please, revise the whole document and correct accordingly.

Response: Thank you for your advice, this part has been corrected accordingly (yellow highlight) in the re-submitted manuscript.

Figure 1. Authors must improve the figure legend, by indicating the meaning of the initials.

Response: Thank you for your advice, figure legend of Figure 1. has been supplemented accordingly (yellow highlight) in the re-submitted manuscript.

Table 1. Please, revise and improve the format of this table. Authors must also provide and appropriate table legend.

Response: Thank you for your advice, Table 1 has been revised accordingly (yellow highlight) in the re-submitted manuscript.

Table 2. Authors must provide and appropriate table legend.

Response: Thank you for your advice, Table 2 has been revised accordingly (yellow highlight) in the re-submitted manuscript.

Figures 2 and 3. Authors must provide and appropriate legend for both figures.

Response: Thank you for your advice, Figures 2 and 3. has been supplemented accordingly (yellow highlight) in the re-submitted manuscript.

The Discussion section needs to be improved by highlighting the physiological relevance of the results obtained and their implications in clinical practice.

Response: Thank you for your advice, this part has been revised accordingly (yellow highlight) in the re-submitted manuscript.

The Conclusion section also needs deep revision and improvement by highlighting most significant results.

Response: Thank you for your advice, this part has been revised accordingly (yellow highlight) in the re-submitted manuscript.

Reviewer 2 Report

In this manuscript, the authors investigated the optimal ranges of some factors associated with a higher CLBR in women undergoing the first in vitro fertilization (IVF) or intracytoplasmic sperm injection (ICSI).

This manuscript is generally well written and the experiments were well designed and performed.However, there are some questions which need to be addressed.

Is there any limits or shortcomings of this study? What should be focus on in future research?

Author Response

In this manuscript, the authors investigated the optimal ranges of some factors associated with a higher CLBR in women undergoing the first in vitro fertilization (IVF) or intracytoplasmic sperm injection (ICSI).

This manuscript is generally well written and the experiments were well designed and performed. However, there are some questions which need to be addressed.

Is there any limits or shortcomings of this study? What should be focus on in future research?

Response: Thank you for your advice, the main limitation in our study was the retrospective nature of itself. And this part has been supplemented accordingly (yellow highlight) in the “Discussion” of the re-submitted manuscript.

Reviewer 3 Report

The aforementioned study provides systematically defined criteria as potential reasons for the cumulative live birth rate. Although this fact is already known, the authors in this paper report for the first time a clear dose-response relationship according to the cumulative live birth rate, which is of exceptional practical importance.

The study, in a concise, sufficiently informative, and of practical importance, tries to explain the long-known importance of AMH in the assessment of ovarian reserve, but also to point out its relative role. The role of FSH in the assessment of ovarian reserve was also discussed.

Nevertheless, the authors found optimal ranges of five factors that were correlated with a higher live birth in the first cycle of oocyte retrieval, which certainly provides a scientific basis for the clinical trial and treatment of IVF.

Disadvantages: The gestational age, i.e. the gestational week as the boundary between abortion and childbirth is not defined.

Author Response

Disadvantages: The gestational age, i.e. the gestational week as the boundary between abortion and childbirth is not defined.

Response: Thank you for your advice, we have supplemented this definition (yellow highlight) in the “Materials and Methods” of the re-submitted manuscript.

Reviewer 4 Report

1.     The authors are advised to reduce the length of the manuscript title

2.     The abstract lacks innovation and appears more like a synopsis. The writers must make it apparent how their work helps close the gap between published work and emerging trends. The research’s objective, key findings, and principal conclusions should be separately stated in the abstract. It must be able to stand alone since an abstract is frequently offered apart from the article.

3.     The design of the introduction is inappropriate. The authors are advised to include some of the latest studies in the introduction section to clarify the work's objective.

4.     The authors should be a little more descriptive about the terms ART, FET, and CLBR

5.     It is advised to outline the article’s structure towards the conclusion of the introduction. Contrasting the current study’s findings with some earlier, related research is advised.

6.     The authors are advised to check the grammatical errors throughout the manuscript.

7.     The conclusion section does not correctly address the important points and future perspectives. The author should add a few important points.

Author Response

  1. The authors are advised to reduce the length of the manuscript title

Response: Thank you for your advice, the manuscript title has been reduced accordingly (yellow highlight) in the re-submitted manuscript.

  1. The abstract lacks innovation and appears more like a synopsis. The writers must make it apparent how their work helps close the gap between published work and emerging trends. The research’s objective, key findings, and principal conclusions should be separately stated in the abstract. It must be able to stand alone since an abstract is frequently offered apart from the article.

Response: Thank you for your advice, the abstract has been revised (yellow highlight) in the re-submitted manuscript based on your advice and the guideline of journal (A single paragraph of about 200 words maximum).

  1. The design of the introduction is inappropriate. The authors are advised to include some of the latest studies in the introduction section to clarify the work's objective.

Response: Thank you for your advice, the introduction has been revised accordingly (yellow highlight) in the re-submitted manuscript.

  1. The authors should be a little more descriptive about the terms ART, FET, and CLBR

Response: Thank you for your advice, the description of terms has been supplemented accordingly (yellow highlight) in the re-submitted manuscript.

  1. It is advised to outline the article’s structure towards the conclusion of the introduction. Contrasting the current study’s findings with some earlier, related research is advised.

Response: Thank you for your advice, the introduction has been revised accordingly (yellow highlight) in the re-submitted manuscript.

  1. The authors are advised to check the grammatical errors throughout the manuscript.

 Response: Thank you for your advice, the grammatical errors has been checked (yellow highlight) in the re-submitted manuscript.

  1. The conclusion section does not correctly address the important points and future perspectives. The author should add a few important points.

 Response: Thank you for your advice, the conclusion section has been revised (yellow highlight) in the re-submitted manuscript.

Round 2

Reviewer 1 Report

Authors introduced all the changes requested by this reviewer and the overall quality of the manuscript has been improved.

Reviewer 4 Report

Accepted